# Quantification and Characterization of CTCs and Clusters in Pancreatic Cancer by Means of the Hough Transform Algorithm

**DOI:** 10.3390/ijms24054278

**Published:** 2023-02-21

**Authors:** Francisco José Calero-Castro, Sheila Pereira, Imán Laga, Paula Villanueva, Gonzalo Suárez-Artacho, Carmen Cepeda-Franco, Patricia de la Cruz-Ojeda, Elena Navarro-Villarán, Sandra Dios-Barbeito, María José Serrano, Cristóbal Fresno, Javier Padillo-Ruiz

**Affiliations:** 1Department of General Surgery, Hospital University Virgen del Rocío/CSIC/University of Seville/IBiS, 41013 Seville, Spain; 2Oncology Surgery, Cell Therapy, and Organ Transplantation Group, Institute of Biomedicine of Seville (IBiS), Virgen del Rocio University Hospital, University of Seville, 41013 Seville, Spain; 3Department of Liquid Biopsy, Genyo, 18016 Granada, Spain; 4Health and Sciences Research Center, Health and Sciences Faculty, Anahuac University, Huixquilucan 52760, Mexico

**Keywords:** circulating tumor cell, cluster, pancreatic cancer, Hough transform

## Abstract

Circulating Tumor Cells (CTCs) are considered a prognostic marker in pancreatic cancer. In this study we present a new approach for counting CTCs and CTC clusters in patients with pancreatic cancer using the Isoflux^TM^ System with the Hough transform algorithm (Hough-Isoflux^TM^). The Hough-Isoflux^TM^ approach is based on the counting of an array of pixels with a nucleus and cytokeratin expression excluding the CD45 signal. Total CTCs including free and CTC clusters were evaluated in healthy donor samples mixed with pancreatic cancer cells (PCCs) and in samples from patients with pancreatic ductal adenocarcinoma (PDAC). The Isoflux^TM^ System with manual counting was used in a blinded manner by three technicians who used Manual-Isoflux^TM^ as a reference. The accuracy of the Hough-Isoflux^TM^ approach for detecting PCC based on counted events was 91.00% [84.50, 93.50] with a PCC recovery rate of 80.75 ± 16.41%. A high correlation between the Hough-Isoflux^TM^ and Manual-Isoflux^TM^ was observed for both free CTCs and for clusters in experimental PCC (R^2^ = 0.993 and R^2^ = 0.902 respectively). However, the correlation rate was better for free CTCs than for clusters in PDAC patient samples (R^2^ = 0.974 and R^2^ = 0.790 respectively). In conclusion, the Hough-Isoflux^TM^ approach showed high accuracy for the detection of circulating pancreatic cancer cells. A better correlation rate was observed between Hough-Isoflux^TM^ approach and with the Manual-Isoflux^TM^ for isolated CTCs than for clusters in PDAC patient samples.

## 1. Introduction

Pancreatic ductal adenocarcinoma (PDAC) is a tumor with poor prognosis due to the high rate of early distant metastasis [1]. Liquid biopsy is a way of obtaining tumor information as a tool for personalized medicine [2,3,4,5]. This technique includes several determinations. One of them is circulating tumor cell (CTC) detection [3,6]. CTCs are released from the primary solid tumor to the bloodstream and may provide clinically relevant information for cancer diagnosis, prognosis, and treatment [7,8]. Therefore, in patients with PDAC, the detection of CTCs and clusters could be really challenging.

Different technologies for CTC isolation from normal hematopoietic cells, such as erythrocytes and leukocytes, have been described [9,10,11,12,13] using differentiating cell parameters, such as physical properties, flow, or elasticity characteristics. Moreover, the differential expression of biological factors, such as putative tumor-associated antigens (TAAs) or simply markers of epithelial vs. mesenchymal/hematopoietic transition have been developed [9]. One of these technologies is the IsoFlux^TM^ System (Fluxion Biosciences, Inc., Alameda, CA, USA). The IsoFlux^TM^ System is a semiautomatic platform that facilitates CTC enrichment by using immunomagnetic positive selection, along with microfluidics [14,15]. This system has been used to successfully isolate CTCs from many cancers [16,17,18,19,20,21,22,23]. Epithelial cell adhesion molecule (EpCAM) is a molecule found on the cell surface that shows high expression levels in solid cancers. In order to detect this molecule, ferromagnetic particles coated with an antibody against EpCAM have been used to enrich CTCs from blood samples. Once enriched, the distinction between CTCs and hematopoietic cells is achieved by means of different approaches, such as the Hoechst 33342 staining, which binds to cell nuclear DNA. Additional antibodies against anti-cytokeratin (CK) and anti-leukocyte antigens, usually CD45, allow for the differentiation of epithelial cells and leukocytes, respectively [9,10,24]. However, the final step in the immunomagnetic separation systems is manual, since a technician must review each subsequent microimage generated to determine if each counted event is Hoechst 33342-positive, CK-positive, and CD45-negative [9].

In addition, it is really important that these methods allow for the detection of the presence of a CTC group, which is labeled as a CTC cluster [9,25,26]. Clusters have been associated with high metastatic potential and a shorter lifespan [26]. A CTC cluster can present a heterogenic cell composition, with platelets, immune cells, or cancer-associated fibroblasts. This cell architecture provides a local microenvironment that shelters clustered CTCs and smooths the path for colonization [27]. Therefore, CTC cluster quantification is a potentially useful parameter to take into consideration as a relevant prognostic marker [9,27].

The Hough transform, described in 1972, is a technique that locates shapes in images. Hough transform is an algorithm that effectively detects straight lines and curves in an image and is widely used in computer processes. Different studies have successfully used the Hough transform to grasp the edge of cells [28,29,30,31,32]. The Hough transform is based on a solid mathematical theory that has been broadly applied in many clinical fields [28,30,31,33,34,35].

Here, we present a novel semi-automatized technique based on the Hough transform and coupled with the Isoflux^TM^ System (Hough-Isoflux^TM^) that allows for the counting of free CTCs and clustered CTCs isolated from whole blood samples of patients with PDAC.

## 2. Results

### 2.1. Experimental Validation

#### 2.1.1. Isolated Pancreatic Cancer Cell (PCC) Evaluation

The median Hough-Isoflux^TM^ sensitivity and accuracy for detecting PCC in the counted events were 89.00% [82.50, 91.00] and 91.00% [84.50, 93.50], respectively. In samples from healthy patients that were taken as negative controls (samples that were not mixed with PCC), only 1 [0, 4] lymphocyte was detected. Figure 1 shows PCC and CTC characterization with specific staining.

Table 1 presents the results of the PCC included in the control groups, as well as the recovery rate obtained with the manual tumor cell counting of the Isoflux^TM^ system (Manual-Isoflux^TM^) and with the Hough transform approach. Although the recovery rate obtained with the Hough-Isoflux^TM^ approach was lower than that with the Manual-Isoflux^TM^, differences were not significant (80.75 ± 16.41% vs. 82.44 ± 9.55%; *p* = 0.801). The correlation coefficients observed between control PCC and both techniques were almost similar. In the Manual-Isoflux^TM^, we obtained a correlation coefficient of R^2^ = 0.988, a slope of 0.696 (95% CI, 0.627–0.764), and an interception of 45.74 (95% CI, −0.553–92.040) (Figure 2a). The Hough-Isoflux^TM^ approach provided a correlation coefficient of R^2^ = 0.974, a slope of 0.649 (95% CI, 0.553–0.744), and an interception of 44.380 (95% CI, −20.010–108.800) (Figure 2b). A high correlation between both the Manual-Isoflux^TM^ and Hough-Isoflux^TM^ techniques was observed obtaining an R^2^ = 0.991, a slope of 0.935 (95% CI 0.857, 1.014), and an interception of 0.614 (95% CI, −39.180–40.410) (Figure 2c).

#### 2.1.2. Cluster Evaluation

The results of cluster, clustered-PCC, and free-PCC counting are presented in Table 2. There were no significant differences between techniques for detecting clusters with an overall R^2^ = 0.902 correlation rate, a slope of 0.628 (95% CI, 0.443–0.8131), and interception of −0.056 (95% CI, −2.629–2.513) (Figure 3a). However, some significant differences were observed for PCC clusters with moderate and high tumor cell concentrations (300–1000 PCCs). The correlation coefficient observed between Manual-Isoflux^TM^ and Hough-Isoflux^TM^ techniques for clustered-PCC counts was R^2^ = 0.940 with a slope of 0.586 (95% CI, 0.454–0.719), and the interception was −0.408 (95% CI, −5.179–4.363) (Figure 3b).

Finally, regarding the remaining free-PCC, both techniques also counted similar PCC numbers with a high correlation rate, as shown in Figure 3c (R^2^ = 0.993; slope of 0.964 [95% CI, 0.893–1.035], and an interception of −1.165 [95% CI, −34.76–32.43]).

### 2.2. Clinical Validation

#### CTCs and Cluster Measurements in Patients with PDAC

Fifty samples from patients with PDAC were evaluated for CTCs and cluster detection. The results are shown in Table 3. There were no significant differences between the Manual-Isoflux^TM^ and the Hough-Isoflux^TM^ approaches.

Figure 4 represents the CTCs counted by the Manual-Isoflux^TM^ System and by the Hough-Isoflux^TM^ System. High correlation coefficients were demonstrated in the total-CTCs (R^2^ = 0.972) and free-CTCs (R^2^ = 0.974). The slopes obtained through the linear regression study were close to 1:0.978 (95% CI, 0.930–1.025) with an interception of 0.677 (95% CI, −24.160, 22.810) in the total-CTCs (Figure 4a). The slope was 1.004 (95% CI, 0.955–1.052) with an interception of 2.470 (95% CI, −18.130–23.070) for the free-CTCs (Figure 4b).

The cluster and clustered-CTC correlation coefficients (R^2^) between both techniques were 0.790 and 0.723, respectively. For clusters (Figure 4c), we obtained a slope of 0.874 (95% CI, 0.743–1.005) and an interception of 3.389 (95% CI, −0.021–6.799). For clustered-CTCs (Figure 4d), a slope of 0.625 (95% CI, 0.513–0.737) and an interception of 8.228 (95% CI, −1.148–17.600) was obtained.

### 2.3. CTC Size and Cluster Size

The Hough-Isoflux^TM^ System allows for the measurement of CTC and cluster sizes. In control samples, the diameter of the free-PCCs was 7.71 ± 0.50 µm, while that of the clustered-PCCs was 11.06 ± 1.17 µm. The cluster size was 383.36 ± 65.59 µm^2^. On the other hand, the diameter of free-CTCs in patient samples was 7.99 ± 1.55 µm, while clustered-PCCs had a diameter of 10.46 ± 1.59 µm. Clusters in patient samples had a surface area of 562.92 ± 291.50 µm^2^.

## 3. Discussion

Over the last few years, it has been demonstrated that the detection of CTCs can be related to patients’ survival. There are several techniques for CTC isolation based on different principles and yields [9,10,11,12,13]. In our case, for CTC isolation, we defined the tumor cell population following the criteria previously published in other works [21,36,37]. EpCAM is one of the most expressed proteins in CTCs and is also overexpressed in tumors and other cells. After the isolation of potential tumoral cells using the EpCAM marker, we identified, via fluorescence microscopy, the actual CTCs as CK-positive, Hoechst 33342-positive, and CD45-negative. This allowed us to avoid the counting of the frequent CD45-positive PMBC contaminants of blood samples [38], as well as red blood cells and platelets, with an absent nucleus. Only nucleated cells with CK marker expression and no CD45 signal were considered the tumor cells of interest. Currently, there are different devices for the isolation of CTCs from enrichment with EpCAM [10,11,12,13,39]. Among them, CellSearch^®^ is currently the only device approved by the FDA for monitoring patients with metastatic breast, colorectal, and prostate cancer [38,39]. However, like Rarecyte^®^, these are semi-automatic devices since they require a technician to assess whether the candidate events are CTCs [11,13]. Previous work has used CellSpotter^®^ Analyzer [10,40] and CellTracks^®^ [11] in combination with different isolation methods to automate the process. It has also been described that Rarecyte^®^ has been used in combination with CyteFinder^®^ [13]. Although this combination of devices and software provides sensitivity, accuracy, and an elevated correlation in the total CTCs detected, supervision by a technician is always necessary. In the present study, the reference measurements have been also made by three experienced technicians. Currently, the new techniques for the isolation and enrichment of CTCs are being based on the use of devices that combine microscopy and cytometry techniques [41,42,43] to allow automated measurements avoiding potential bias. However, these samples cannot be fixed as we did in the present study due to the use of immunofluorescence.

As previously reported [44,45], we chose the EpCAM-Isoflux™ System because this device could isolate more CTCs than CellSearch^®^, and both of them use the same isolation target, EpCAM. Nevertheless, a straight-forward comparison of the CTC isolation performed using the two platforms is challenging, due to the fact that they use different antibody clones and different magnetic beads for the capture process. Thanks to the micrometer-scale beads included in the IsoFlux™ System, the magnetic moment generated is sufficient for capturing even cells with low expression levels of markers [44]. Moreover, many works have reported the good results for the Isoflux^TM^ when isolating CTCs from different cancer types [16,17,18,19,20,21,22,23]. For the counting of isolated events using the Isoflux™, the Isoflux^TM^ Cytation Imager software is frequently used, but no evidence has been found in terms of its sensitivity and accuracy.

Our proposed technique, like many approaches to isolate CTCs, is a semi-automatic method in which user interaction is only required for threshold specification and noise removal. Previous works use different algorithms to detect cells, such as edge detection [46,47,48] or concavity detection. Edge detection can be performed based on Laplacian, Gaussian, or Canny algorithms.

Therefore, this paper presents a study in which we relied on the Hough transform- Isoflux^TM^ System to perform the semi-automatic counting of CTCs detected based on EpCAM staining. First, when we evaluated the coefficient of variation (CV) of the novel approach, we found that the CV measured with the Hough-Isoflux^TM^ System did not exceed 2% in CTC detection. This indicates homogeneity in the counting and minimal variation from one count to another based on the same sample.

Since Hough transform is an image analysis technique, it can potentially be coupled to other platforms that isolate CTCs, such as the CellSearch^®^ or CellSpotter^®^ Analyzer, which provide a final image to be analyzed. The markers used can differ between the different approaches, but in general, they should be a molecule or a combination of molecules expressed only in cancer cells. There are specific markers that once detected in blood cells for cancer patients, if the cell is CD45 negative, allow us to infer that we are dealing with a foreign cell and probably a cancerous one. These markers are divided into three groups: epithelial markers (such as EpCAM, cytokeratins, or E-cadherin), mesenchymal markers (such as vimentin, N-cadherin, or Twist1), and stem markers (such as CD44 or ALDH1) [49]. Moreover, some works mentioned more specific markers, such as BRAF^V600E^ and PD-L1 [14]. These markers can be considered in the immunofluorescent step to detect them with a microscope, taking into consideration the wavelength of each marker.

Regarding cancer cell recovery, similar to other approaches [14], the Hough-Isoflux^TM^ System reported an 80.75% rate of pancreatic cell cancer recovery. Moreover, we found a high sensitivity (89.00%) and accuracy (91.00%) for CTC detection, similar to those with other analyzers, such as the CellSpotter^®^ [10], which showed a sensitivity of 85% and an accuracy of 99.7%. Although sensitivity and accuracy showed good results, they are not perfect. In that line of thought, some authors, such as Chen et al. [50], improved their results by combining CTC with CA19-9.

Even though the available literature provides several detection sensibilities, those values are not comparable to those of our study. The reason for that is that we used a high-yield enrichment technique, followed by an automated method of enumeration. This led to the detection of CTC in 100% of the studied samples from patients with early pancreatic cancer.

When the correlation between both techniques was evaluated, high linearity was observed between Manual-Isoflux^TM^ System counting and Hough transform-Isoflux^TM^ System counting. This was observed in both total and free-PCCs in blood from healthy donors, as well as in total and in free-CTCs from patients with pancreatic cancer.

Since clusters are related to tumor metastasis [9,25,26], the determination of clusters is crucial for clinical purposes. In our study, there was a high correlation between both techniques analyzed for cluster detection in experimental PCC, whereas a lesser correlation was found in samples from patients with PDAC. This could be explained because we have included restricted parameters in the software to avoid counting artifacts in the sample. In this sense, we thought that it would be better to improve the accuracy instead to increase the false positives. Moreover, another explanation for the lower correlation coefficients between both the Manual-Isoflux^TM^ System and Hough Transform-Isoflux^TM^ System may be related to the binary mask generated during the data processing after the logical operations performed with each color channel. This binary mask is obtained after the logical multiplication of the blue (Hoechst 33342+) and green (CK+) channels, and with the result, a logical operation is performed with the red channel (CD45+) to discard the cells with CD45 expression. The implementation of this mask is a crucial step for the high sensibility and accuracy in the counts of CTCs, which have a well-defined rounded morphology. Clusters, on the other hand, show irregular shapes with two or more CTCs within them, which often result in overlapping CTCs that are more difficult to identify based on the Hough transform. A potential approach to improve the correlation coefficients could be to increase the sensitivity factor to find overlapping circles in an irregular shape. However, this factor may count events that are not CTCs or count a CTC more than once. Thus, this interesting point needs further research.

Using the Hough transform-Isoflux^TM^ System, we could determine the CTC and cluster size. However, these measurements are an estimation because we measured the correlation between Hoechst 33342 and CK (Figure 1). CK labeled the cytoplasm, whereas Hoechst 33342 labeled the cell nucleus. Therefore, in some cases, CK was more representative than the correspondence in determining the cell size.

A limitation of our method for this specific clinical approach is that it requires more memory than that in other fields since several images are needed due to the generation of masks. This process can produce images larger than 7000 × 7000 pixels. Therefore, the computational cost of the technique is high. Additionally, Hough transform returns a huge matrix with the cell center and the counted cell size. Nevertheless, according to our experience, Hough transform is simple to implement and can generate results in real-time, which is highly suitable for CTC counting for clinical applications.

In conclusion, the Hough-Isoflux^TM^ System is a good approach for counting CTCs, clusters, and clustered CTCs based on their identification as nucleated cells with EpCAM expression, as well as CK-positive and CD45-negative cells, once they have been isolated with the Isoflux^TM^ System. Moreover, the strategy hereby presented might allow for the counting of other cells of interest by processing image channels depending on the marker expression profile of the cells.

## 4. Materials and Methods

### 4.1. Collection of the Samples

In total, 50 samples from 9 patients with adenocarcinoma in the head of the pancreas were enrolled in this study. Following the National Comprehensive Cancer Network (NCCN) guidelines V1.2013 of Pancreatic Cancer and the American Joint Committee on Cancer (AJCC) TNM Staging of Pancreatic Cancer, the study included 33.33% of patients with stage I (11.11% IA, 22.22% IB), 44.44% with stage II (33.33% IIA, 11.11% IIB), and 22.22% of patients with stage III disease after a surgical pathology study.

Blood samples from two healthy donor volunteers with nonmalignant diseases were used as a control group. Tubes with EDTA were used to collect blood samples, which were stored at room temperature and processed before 24 h after collection.

### 4.2. Cell Culture

The pancreatic cancer cell line PANC-1 (ATCC^®^ CRL-1469, Manassas, VA, USA) was cultured in Dulbecco’s Modified Eagle’s Medium (DMEM, Lonza BE12-604F, Basel, Switzerland), previously supplemented with 10% fetal bovine serum (F7524, Sigma-Aldrich, St. Louis, MI, USA) and 1% Penicillin/Streptomycin (15140122, GIBCO, Waltham, MA, USA). A humidified incubator at 37 °C with 5% CO_2_ was used to grow the cell cultures. Once seeded, cell cultures were washed with phosphate-buffered saline (PBS, H3BE17-516F, Lonza) and trypsinized using Trypsin-EDTA at 0.05% (25300-062, Gibco, Waltham, MA, USA). Pancreatic cancer cells (PCCs) were used to calibrate the method.

### 4.3. Sample Preparation for Recovery, Accuracy, Sensitivity, and Linearity of PCC Detection

Two different groups were used for accuracy, sensitivity, and linearity experiments. Seven-milliliter samples of blood were collected from a donor with no physical affection (n = 3) and were prepared and mixed with approximately 100, 300, and 1000 PCCs. In the second group, for false-positive assays, 7 mL samples of blood collected from a healthy donor were processed in EDTA tubes. All samples were counted by three blinded expert technicians using the Hough transform algorithm. We defined the Manual-Isoflux^TM^ recovery as the number of manually counted PCCs per control PCC and Hough-Isoflux^TM^ System recovery as the number of Hough transform-Isoflux^TM^ System-counted PCCs per control PCC. Accuracy was defined as well-counted PCCs per Hough transform-Isoflux^TM^ System-counted PCC, whereas sensitivity was defined as well-counted PCCs per control PCCs.

### 4.4. Reproducibility of CTC Measurements between Duplicate Samples and Multiple Operators

Fifty samples were collected from patients with pancreatic cancer and were processed as we describe in the following section. All samples were counted by three blinded expert technicians (Manual-Isoflux^TM^ System in text) and the Hough transform algorithm- Isoflux^TM^ System after CTC isolation and enrichment was performed as described in Section 4.5.

### 4.5. Isolation of Circulating Tumor Cells

To enrich blood samples from peripheral mononuclear blood cells, gradient centrifugation with Histopaque^®^-1119 was used, while the IsoFlux platform was applied for CTC and PCC isolation. The IsoFlux™ System takes advantage of the magnetic moment generated by the micrometer-scale beads that it contains to capture cells, even those with low expression levels of target markers [27]. Moreover, this is an automatic process that increases the capture rate of PCCs and CTCs. For CTC enrichment, the Isoflux™ Epithelial to Mesenchymal Transitions Circulating Tumor Cell Enrichment Kit (EMT Enrichment Kit, Izasa, Catalog N.910-0106, San Diego, CA, USA) was used, where beads targeting epithelial and mesenchymal markers were conjugated with four different antibodies. For the detection of epithelial cells, anti-EpCAM (a highly expressed molecule in solid cancers) and anti-EGFR antibodies are used in the kit, while anti-N-Cadherin and anti-Vimentin were used to target mesenchymal markers. The cells suspended in IsoFlux Binding Buffer were combined with the immunomagnetic beads conjugated with the above-mentioned antibodies and incubated for 90 min at 4 °C, as instructed in the Fluxion protocol. The cell and bead mixture was loaded into the microfluidic cartridge and processed with the Isoflux™ System. Then, in the isolation zone inside the instrument, the cells were exposed to a magnetic field towards which the target cells are attracted with CTC beads attached. The target cells were collected on a removable disk. Once the samples had been processed, fluorescent reagents (Isoflux™ Circulating Tumor Cell Enumeration Kit Izasa Catalog N.910-0093, Fluxion, San Diego, CA, USA) were used to fix and stain the enriched cells. The following antibodies were used: anti-CK-fluorescein isothiocyanate (FITC) conjugated, which binds specifically to CK, an intracellular protein typical of epithelial cells; anti-CD45/Indocarbocyanine (Cy3), which is expressed only in leukocytes; and Hoechst 33342, a nucleic acid marker that emits blue fluorescence once it binds to the minor grove of the DNA, where AT-rich regions can be found. For the acquisition of fluorescent images of the stained samples, an Olympus BX-61 Direct Fluorescence microscope was used. In light of the above, CTCs or PCCs are defined as CK+/CD45-/nucleated and morphologically intact cells.

The samples that were not isolated with the IsoFlux were stained with fluorescent reagents (Isoflux™ Circulating Tumor Cell Enumeration Kit Izasa Catalog N.910-0093, Fluxion, San Diego, CA, USA).

### 4.6. Software Counting Development

#### 4.6.1. Image Pre-Processing

The sample images stained with Hoechst 33342, CK, and CD45 were given as input to the system to be analyzed. The images were collected from normal donors and cancer patients. We used the blue channel (Hoechst 33342), the green channel (CK), and the red channel (CD45). Every track was converted to grayscale by eliminating the hue and saturation element and retaining luminance. 

#### 4.6.2. Image Segmentation

The images were converted to binary form using the global Otsu’s threshold to minimize the intra-class variance. Pixels with a grayscale value higher than the threshold calculated by Otsu’s algorithm were marked as one, while those that did not exceed the threshold were marked as zero. The histogram allowed us to select the threshold if Otsu’s value is not high enough. As previously described, CTCs are CK+/CD45-/nucleated cells. Therefore, image pixels that show Hoechst 33342+, CK+, and CD45- are considered CTCs. A logical multiplication between the threshold values of blue and green channels (Hoechst 33342 and CK, respectively) was applied to the images. Finally, we removed the thresholding pixels of channel red (CD45). The goal was to keep the pixels dyed blue and green but not red; for this, binarization of the three channels was carried out. The artifacts from the obtained mask were removed using Matlab tools (such as imcrop or roipoly, where the user can select one component to delete). Considering that the resolution of the images was 0.75 pixel/µm and that the size of the CTCs of patients with pancreatic adenocarcinoma is not defined, the size of the CTCs was considered between 5 and 25 µm, according to studies in other tumors [10]. Image components with a size between 10 and 400 pixels were filtered. If the component size exceeded 400 pixels, it was introduced in a new mask, where the CTCs would be counted in clusters. Another filter was circularity, which must be greater than 0.6; if it was less than that value, the components were transferred to the cluster counting mask (Figure 1).

#### 4.6.3. Image Post-Processing

We multiplied the masks obtained with the green channel (grayscale) to facilitate algorithm counting based on the CK intensities. Finally, using the Hough transform through the imfindcircles Matlab function, we measured the components in the initial mask and the clustered CTC mask. This function returned the pixel where the cell centers were and the size of the event. If an element had only the associated center of a circumference of the Hough transform, it was taken as a free-CTC, while if any of the components had two or more pixels as the center of a Hough transform circumference, it would be taken as a cluster with as many cells as centers that had been detected. This methodology was validated by checking the counted event from healthy donors with the Hough transform algorithm. The software determined the number of free-CTCs, clusters, clustered-CTCs, and total CTCs (free and clustered-CTCs). The software showed the average and median size of free-CTCs and clustered-CTCs based on the radius. At the same time, the cluster surface was measured based on the pixel number and the image resolution. An analysis of the measurement’s intra-assay coefficient of variation (CV) was provided. Thereby, mean values of CV were 0.42% ± 0.33 for the total-CTCs, 0.30% ± 0.37 for the free-CTCs, 0.72% ± 1.25 for the clusters, and finally, 1.41% ± 1.22 for clustered-CTCs.

### 4.7. Statistics

An independent student’s t-test was performed to compare both counting approaches. When quantitative variables did not show a normal distribution, the Mann-Whitney U test was used for their analysis. To analyze the relationship between counted PCC and CTC techniques, Spearman’s rank correlation coefficient (R^2^) was used. The required significance was *p* < 0.05. The software used to perform all statistical calculations was the following: IBM Corp. released 2020. IBM SPSS Statistics for Windows, Version 27.0. Armonk, NY: IBM Corp and GraphPad Prism 9.0.0.

## 5. Conclusions

The Hough transform- Isoflux^TM^ System is a good approach for counting CTCs, clusters, and clustered-CTCs, defined as EpCAM nucleated cells, positive for CK, but CD45-negative, after isolation with the Isoflux^TM^ System.

## 6. Patents

Given the inventive characteristics of this innovation, this work has been protected as a patent and utility model by the Spanish Patent and Trademark Office with the ID 2112140033157 in Safe Creative as “The use of the Hough’s transform algorithm to count Circulating Tumour Cells (CTCs)”.

## Figures and Tables

**Figure 1 ijms-24-04278-f001:**
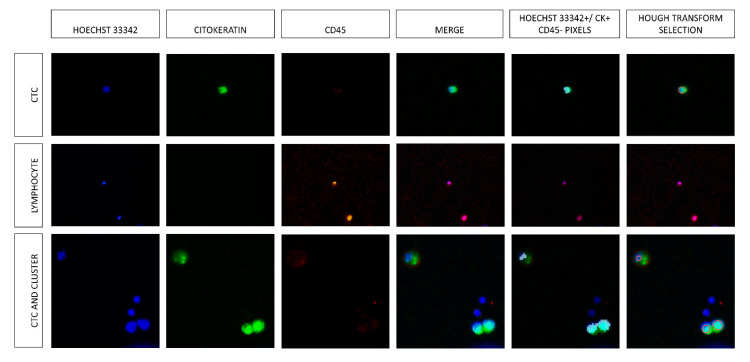
Row one shows the CTC staining characterized by Hoechst 33342+/CK+/CD45-. The fifth image in the first row shows the selection of pixels with CTC staining, and the last photo shows a CTC surrounded by a red circumference counted by the Hough transform. Row two shows the staining of some lymphocytes. Row three shows a free-CTC and a cluster consisting of two cells. Columns five and six show that the Hough transform algorithm did not count the CTCs if the element had no CK staining.

**Figure 2 ijms-24-04278-f002:**
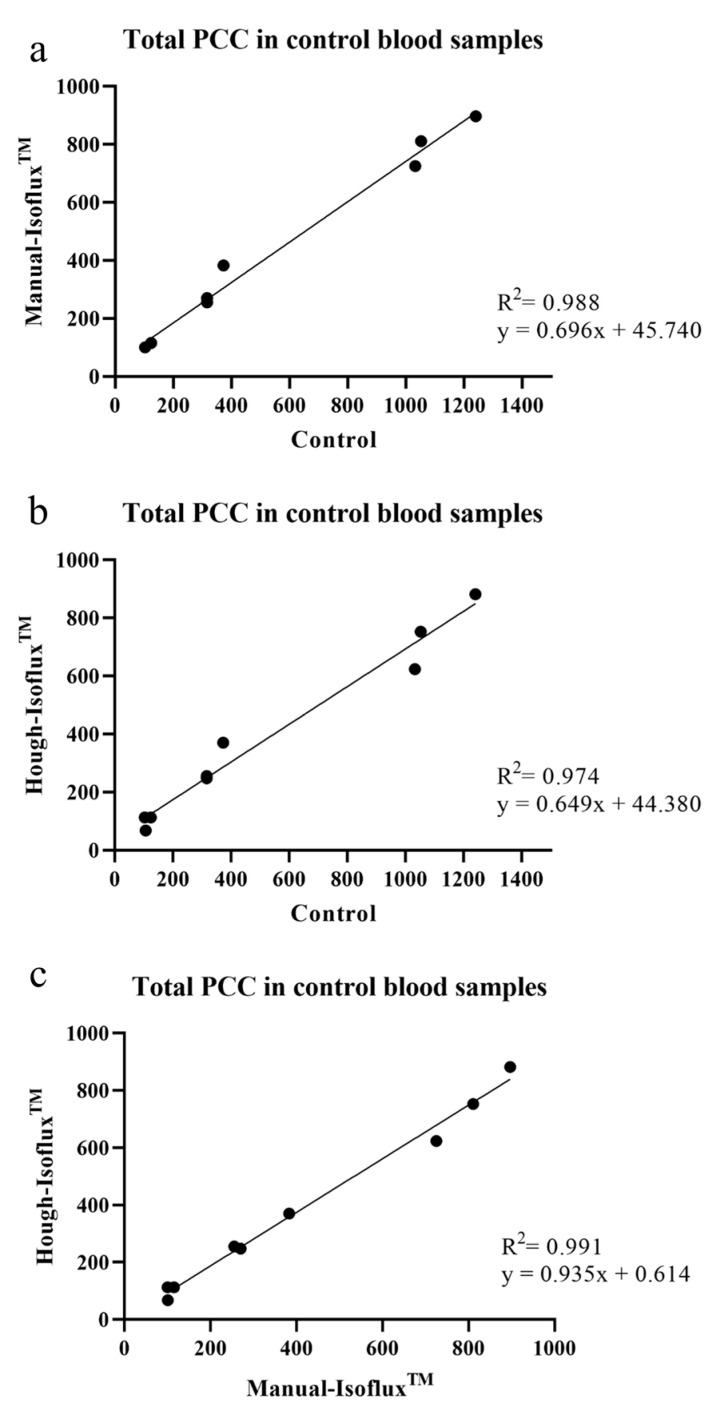
(**a**). Correlation of total PCCs in control blood samples between control and Manual-Isoflux^TM^ System. (**b**). Total PCCs in control blood samples between control and Hough-Isoflux^TM^ System. (**c**). Total PCCs in control blood samples between Manual-Isoflux^TM^ System and Hough-Isoflux^TM^ System.

**Figure 3 ijms-24-04278-f003:**
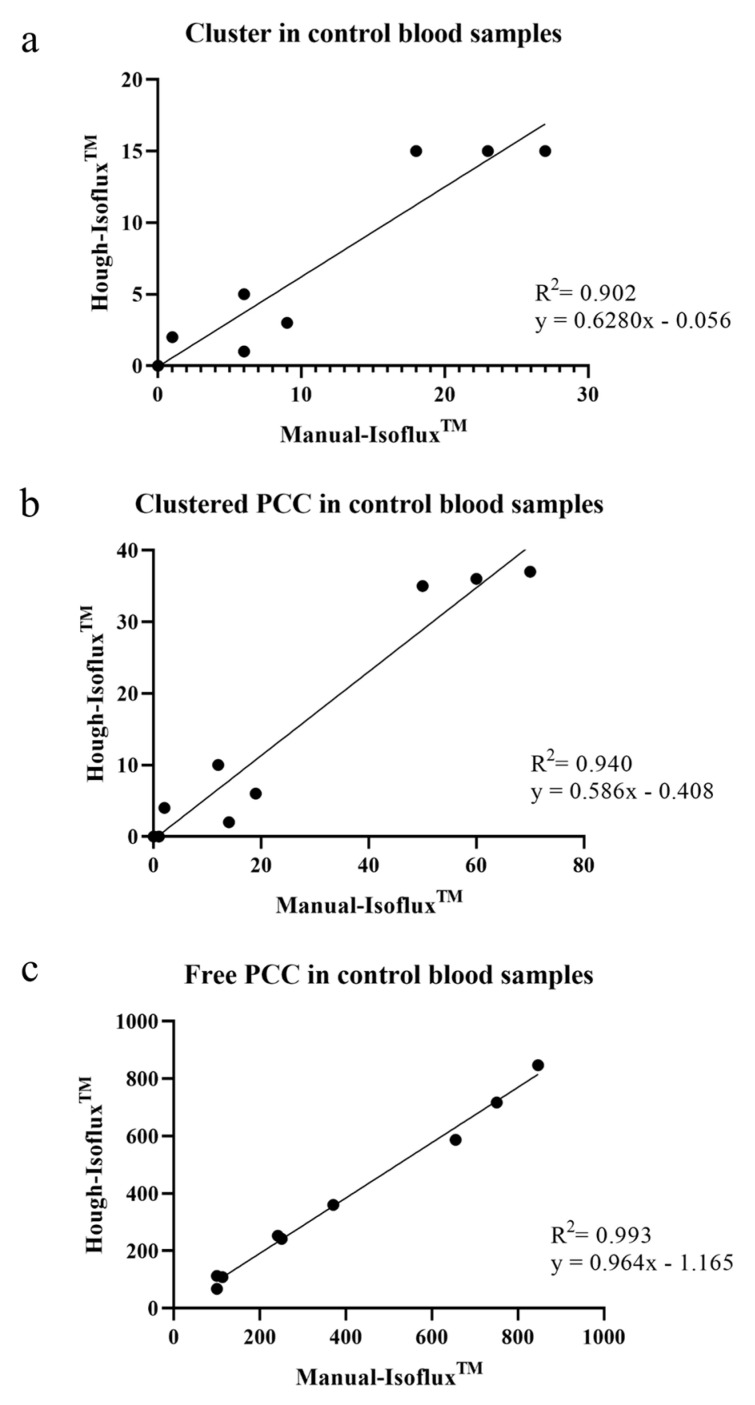
(**a**). Cluster measurement in control blood samples: Manual-Isoflux^TM^ System vs. Hough-Isoflux^TM^ System. (**b**). Clustered-PCC measurement in control blood samples: Manual-Isoflux^TM^ System vs. Hough-Isoflux^TM^ System. (**c**). Free PCC measurement in control blood samples: Manual-Isoflux^TM^ System vs. Hough-Isoflux^TM^ System.

**Figure 4 ijms-24-04278-f004:**
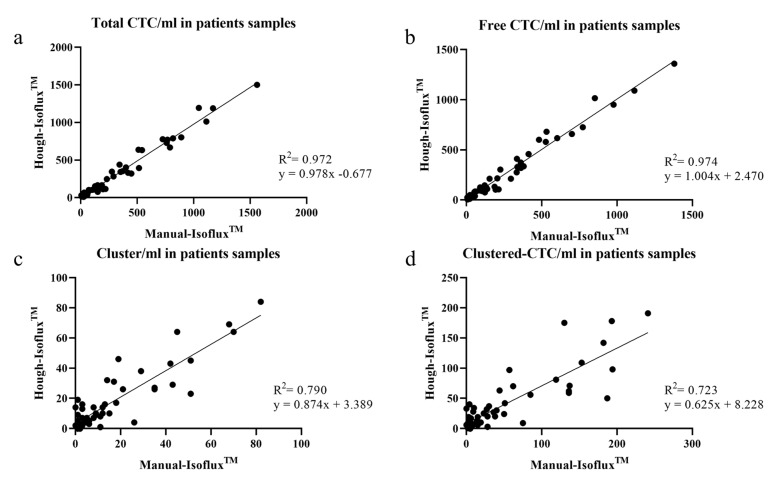
Correlation plot of CTC/mL in pancreatic cancer patient samples counted with the Manual-Isoflux^TM^ System and with the Hough-Isoflux^TM^ System. (**a**). Total-CTC correlation. (**b**). Free-CTC correlation. (**c**). Cluster correlation. (**d**). Clustered-CTC correlation.

**Table 1 ijms-24-04278-t001:** Total counted PCCs with the Manual-Isoflux^TM^ System and Hough-Isoflux^TM^ System. Data are presented with the mean and S.D.

Control Density PCC	Counted Cells (n)			Recovery from Control (%)		
	Manual-Isoflux^TM^	Hough Transform-Isoflux^TM^	*p*-Value	Manual-Isoflux^TM^	Hough Transform -Isoflux^TM^	*p*-Value
No added PCC	3.67 ± 2.60	6.40 ± 4.34	0.263	-	-	-
100	105.33 ± 8.39	98.00 ± 25.98	1.000	89.92 ± 10.98	88.43 ± 22.75	0.924
300	302.67 ± 69.97	291.00 ± 68.51	0.847	84.04 ± 2.83	86.17 ± 11.45	0.782
1000	811.00 ± 86.00	753.00 ± 129.00	0.552	73.16 ± 3.50	67.66 ± 6.29	0.256

**Table 2 ijms-24-04278-t002:** Breakdown of clusters, clustered-PCCs, and free-PCCs. Data are presented with the mean and SD.

Control Density PCC	Clusters (n)			Clustered-PCCs (n)			Free-PCCs (n)		
	Manual-Isoflux^TM^	Hough Transform-Isoflux^TM^	*p* Value	Manual-Isoflux ^TM^	Hough Transform-Isoflux^TM^	*p* Value	Manual-Isoflux^TM^	Hough Transform-Isoflux^TM^	*p* Value
No added PCC	0	0	-	0	0	-	3.67 ± 2.60	6.40 ± 4.34	0.263
100	0.33 ± 0.58	0.67 ± 1.15	1.000	0.67 ± 1.15	1.33 ± 2.31	1.000	104.67 ± 7.23	96.67 ± 24.91	0.621
300	7.00 ± 1.73	3.00 ± 2.00	0.100	15.00 ± 3.61	6.00 ± 4.00	0.044	287.67 ± 72.34	285.00 ± 65.18	0.964
1000	22.67 ± 4.51	15.00 ± 00	0.042	60.00 ± 10.00	36.00 ± 1.00	0.014	751.00 ± 96.00	717.00 ± 130.00	0.736

**Table 3 ijms-24-04278-t003:** CTCs were detected by the Manual-Isoflux^TM^ System and by the Hough transform-Isoflux^TM^ System algorithm in patients with pancreatic cancer.

Group	Manual-Isoflux^TM^	Hough-Isoflux^TM^	*p*-Value
Total-CTCs/mL	192.00 (70, 513)	152 (83.25, 487.50)	0.912
Cluster/mL	8.50 (2, 22.50)	10 (3.75, 26.25)	0.437
Clustered-CTCs/mL	25 (5.75, 77.50)	24.50 (8.75, 59.75)	0.942
Free-CTCs/mL	146.00 (54.50, 388.25)	124.5 (72.50, 422)	0.994

## Data Availability

The data presented in this study are available upon request from the corresponding author.

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
