# Peer review of "Quantification and Characterization of CTCs and Clusters in Pancreatic Cancer by Means of the Hough Transform Algorithm"

_ijms, 2023, doi:10.3390/ijms24054278_

Round 1

Reviewer 1 Report

This is an interesting article about the detection of CTC and CTC clusters in PDAC cases using a novel semi-automated Hough-IsofluxTM approach that was compared to the standard Isoflux system with manual counting. This is an important contribution to the field of CTC and pancreas cancer research as the detection of CTC in PDAC has proved difficult with other technologies due to their heterogeneity, particularly when positive selection markers are used. There is also an analysis of CTC clusters, which is also an important prognostic marker in other cancer types.

Minor comments:

The stage of the tumor of the patients included in the study is not mentioned in the methods. I assume there are surgical cases due to the affiliation of the authors. Are there any metastatic patients? I would think so, as the CTC count is quite high for patients with localized disease.

The sensitivity and specificity of the device is good, although not perfect. Have the authors considered combining this method with other blood-based markers such as CA19-9 or CEA to improve sensitivity/specificity?

Was there any confirmation that the cell population detected were actually tumor cells, e.g. KRAS mutation testing.

Author Response

Dear reviewer,

Thank you very much for your comments and suggestions, which we have followed to improve our paper. Please, appended below you will find a detailed response to your considerations. We have answered all the suggested comments and we believe the manuscript has been greatly improved as a result. The changes have been highlighted in the manuscript in yellow. We hope you find them satisfactory to recommend the publication of the work.

Point 1: This is an interesting article about the detection of CTC and CTC clusters in PDAC cases using a novel semi-automated Hough-IsofluxTM approach that was compared to the standard Isoflux system with manual counting. This is an important contribution to the field of CTC and pancreas cancer research as the detection of CTC in PDAC has proved difficult with other technologies due to their heterogeneity, particularly when positive selection markers are used. There is also an analysis of CTC clusters, which is also an important prognostic marker in other cancer types. Minor comments:

The stage of the tumor of the patients included in the study is not mentioned in the methods. I assume there are surgical cases due to the affiliation of the authors. Are there any metastatic patients? I would think so, as the CTC count is quite high for patients with localized disease.

Response 1: Thank you very much for your comment. Metastatic patients were not included in this study. Following the National Comprehensive Cancer Network (NCCN) guidelines V1.2013 of Pancreatic Cancer and the American Joint Committee on Cancer (AJCC) TNM Staging of Pancreatic Cancer, the study included 33.33% of patients with stage I (11.11% IA, 22.22%  IB); 44.44% stage II (33.33% IIA, 11.11% IIB) and  22.22% of patients with stage III of disease after surgical pathology study.

The high CTC count could be due to the use of the EpCAM-Isoflux™ System because the IsoFlux™ System includes micrometer-scale beads that generate a magnetic moment high enough for capturing even cells with low expression levels of markers. This is consistent with the findings observed in other studies.

  • Agerbæk MØ, Bang-Christensen SR, Yang M-H, Clausen TM, Pereira MA, Sharma S, et al. The VAR2CSA malaria protein efficiently retrieves circulating tumor cells in an EpCAM-independent manner. Nat Commun. 2018 Aug;9(1):3279.
  • Amado V, González-Rubio S, Zamora J, Alejandre R, Espejo-Cruz ML, Linares C, Sánchez-Frías M, García-Jurado G, Montero JL, Ciria R, Rodríguez-Perálvarez M, Ferrín G, De la Mata M. Clearance of Circulating Tumor Cells in Patients with Hepatocellular Carcinoma Undergoing Surgical Resection or Liver Transplantation. Cancers (Basel). 2021 May 19;13(10):2476. doi: 10.3390/cancers13102476. PMID: 34069569; PMCID: PMC8160727.
  • Ramirez P, Sáenz L, Cascales-Campos PA, González Sánchez MR, Llàcer-Millán E, Sánchez-Lorencio MI, Díaz-Rubio E, De La Orden V, Mediero-Valeros B, Navarro JL, Revilla Nuin B, Baroja-Mazo A, Noguera-Velasco JA, Sánchez BF, de la Peña J, Pons-Miñano JA, Sánchez-Bueno F, Robles-Campos R, Parrilla P. Oncological Evaluation by Positron-emission Tomography, Circulating Tumor Cells and Alpha Fetoprotein in Patients With Hepatocellular Carcinoma on the Waiting List for Liver Transplantation. Transplant Proc. 2016 Nov;48(9):2962-2965. doi: 10.1016/j.transproceed.2016.07.035. PMID: 27932119.

We have added these comments to the Methods and Discussion sections, as well as new related references.

Point 2: The sensitivity and specificity of the device is good, although not perfect. Have the authors considered combining this method with other blood-based markers such as CA19-9 or CEA to improve sensitivity/specificity?

Response 2: Thank you very much for this suggestion. You are right when you say that sensitivity and specificity are not perfect. In that line of thought, some authors as Chen et al. improved their results by combining CTC with CA19-9. Your consideration will be kept in mind for future works and we will add this potential improvement in the discussion section.

  • Chen J, Wang H, Zhou L, Liu Z, Tan X. A combination of circulating tumor cells and CA199 improves the diagnosis of pancreatic cancer. J Clin Lab Anal. 2022 May;36(5):e24341. doi: 10.1002/jcla.24341. Epub 2022 Mar 25. PMID: 35334495; PMCID: PMC9102772.

We included this recommendation to the Discussion section, as well as a new related reference.

Point 3: Was there any confirmation that the cell population detected were actually tumor cells, e.g. KRAS mutation testing.

Response 3: Thank you very much for your observation. Although KRAS mutation has not been tested in our study, we defined the tumor cell population following the criteria previously published in other works, cited below. Briefly, we used EpCAM, which is one of the most expressed proteins in CTCs and it is also overexpressed in tumors and other cells. After the isolation of potential tumoral cells using the EpCAM marker, we identified by fluorescence microscopy the actual CTCs as CK positive, Hoechst 33342 positive, and CD45 negative, avoiding this way the count of the frequent CD45 positive PMBC contaminants of blood samples, as well as red blood cells and platelets, with an absent nucleus. Only nucleated cells with CK marker expression and no CD45 signal were considered the tumor cells of interest.  

  • Vasseur A, Kiavue N, Bidard F-C, Pierga J-Y, Cabel L. Clinical utility of circulating tumor cells: an update. Mol Oncol [Internet]. 2021;15(6):1647–66. Available from: https://febs.onlinelibrary.wiley.com/doi/abs/10.1002/1878-0261.12869
  • Parkinson, D.R., Dracopoli, N., Petty, B.G. et al. Considerations in the development of circulating tumor cell technology for clinical use. J Transl Med 10, 138 (2012). https://doi.org/10.1186/1479-5876-10-138
  • Hendricks A, Dall K, Brandt B, Geisen R, Röder C, Schafmayer C, Becker T, Hinz S, Sebens S. Longitudinal Analysis of Circulating Tumor Cells in Colorectal Cancer Patients by a Cytological and Molecular Approach: Feasibility and Clinical Application. Front Oncol. 2021 Jun 28;11:646885. doi: 10.3389/fonc.2021.646885. PMID: 34262858; PMCID: PMC8273730.
  • Vilhav C, Engström C, Naredi P, Novotny A, Bourghardt-Fagman J, Iresjö BM, Asting AG, Lundholm K. Fractional uptake of circulating tumor cells into liver-lung compartments during curative resection of periampullary cancer. Oncol Lett. 2018 Nov;16(5):6331-6338. doi: 10.3892/ol.2018.9435. Epub 2018 Sep 12. PMID: 30405768; PMCID: PMC6202519.

Your comments and new references have been added to the Discussion section.

Reviewer 2 Report

This manuscript is well written and structured; introduces a combination of CTC isolation platform (Isoflux) with an algorithm-Hough transform for semi-automation of CTC and CTC cluster counts, which is of great interest for some medical researchers with rare cell identification background. However in the discussion part, 1) the authors could also discuss how Hough transformation could be used together with other CTC platforms and other potential markers (what characteristics those markers should have?) in a much broader context for more generalised applications. 2) I would also appreciate if the authors could explain more about why correlation coefficients between manual and Hough techniques were relatively poor in assessing clusters than free CTCs, is this because of irregular shape or larger area size of clusters?  How can this be improved? 3) the limitations of Hough transform algorithm in the applications

Author Response

Dear reviewer,

Thank you very much for your comments and suggestions, which we have followed to improve our paper. Please, appended below you will find a detailed response to your considerations. We have answered all the suggested comments and we believe the manuscript has been greatly improved as a result. The changes have been highlighted in the manuscript in yellow. We hope you find them satisfactory to recommend the publication of the work.

Point 1: This manuscript is well written and structured; introduces a combination of CTC isolation platform (Isoflux) with an algorithm-Hough transform for semi-automation of CTC and CTC cluster counts, which is of great interest for some medical researchers with rare cell identification background. However in the discussion part:

The authors could also discuss how Hough transformation could be used together with other CTC platforms and other potential markers (what characteristics those markers should have?) in a much broader context for more generalised applications.

Response 1: Thank you for your remark. Hough transform is a technique used in digital image analysis that allows the detection of edges and the isolation of shapes within an image. For this reason, the Hough transform can be applied to multiple areas of interest, medical or not, and coupled to different platforms in which a final image is obtained and digital analysis is required.

Focusing on its clinical potential, the Hough transform can be used for erythrocyte counting, for the identification and counting of cones in the human retina, or for the detection and segmentation of anatomical structures in noisy medical images, such as femoral heads.

Since Hough transform is an image analysis technique, potentially it can be coupled to other platforms that isolate CTCs, such as CellSearch® or CellSpotter® Analyzer, and provide a final image to be analyzed. The markers used can differ between the different approaches, but in general, they should be a molecule or a combination of molecules expressed only in cancer cells. There are specific markers that once detected in blood cells for cancer patients, if the cell is CD45 negative, allow us to infer that we are dealing with a foreign cell, and probably a cancerous one. These markers are divided into three groups: epithelial markers (such as EpCAM, cytokeratins, or E-cadherin), mesenchymal markers (such as vimentin, N-cadherin or Twist1), and stem markers (such as CD44 or ALDH1). Moreover, some works mentioned more specific markers such as BRAFV600E and PD-L1. These markers can be considered in the immunofluorescent step to detect them at the microscope, taking into consideration the wavelength of each marker.

  • Smith, R., Najarian, K. & Ward, K. A hierarchical method based on active shape models and directed Hough transform for segmentation of noisy biomedical images; application in segmentation of pelvic X-ray images. BMC Med Inform Decis Mak 9 (Suppl 1), S2 (2009). https://doi.org/10.1186/1472-6947-9-S1-S2
  • Barriere G, Fici P, Gallerani G, Fabbri F, Zoli W, Rigaud M. Circulating tumor cells and epithelial, mesenchymal and stemness markers: characterization of cell subpopulations. Ann Transl Med. 2014 Nov;2(11):109.
  • Ruiz-Rodríguez AJ, Molina-Vallejo MP, Aznar-Peralta I, González Puga C, Cañas García I, González E, et al. Deep Phenotypic Characterisation of CTCs by Combination of Microfluidic Isolation (IsoFlux) and Imaging Flow Cytometry (ImageStream). Cancers (Basel) [Internet]. 2021;13(24). Available from: https://www.mdpi.com/2072-6694/13/24/6386

These comments and new references have been added to the Introduction and Discussion sections.

Point 2: I would also appreciate if the authors could explain more about why correlation coefficients between manual and Hough techniques were relatively poor in assessing clusters than free CTCs, is this because of irregular shape or larger area size of clusters?  How can this be improved?

Response 2: This is a good observation. The poor correlation coefficients between both techniques may be related to the binary mask generated during the data processing after the logical operations performed with each color channel. This binary mask is generated after the logical addition of the blue (Hoechst 33342+) and green (CK+) channels, and with the result, a logical multiplication is performed with the red channel (CD45+). The implementation of this mask is a crucial step for the high sensibility and specificity in the count of the CTCs, which have a well-defined rounded morphology. Clusters, on the other hand, show irregular shapes with 2 or more CTCs within them, which often result in overlapped CTCs that are more difficult to identify for the Hough transform.

A potential approach to improve the correlation coefficients could be to increase the sensitivity factor to find overlapping circles in an irregular shape. However, this factor may count events that are not CTC or count a CTC more than once. Thus, this interesting point needs further research by our team.

Your comments have been added to the Discussion section.

Point 3: Limitations of the Hough transform algorithm in the applications

Response 3: This is a very interesting point. Although some limitations have been described in the discussion, the 2 main limitations regarding clinical application are:

  • The technique, like many approaches to isolate CTCs, is a semi-automatic method in which user interaction is required for threshold specification and noise removal.
  • The Hough transform for this specific clinical approach requires more memory than in other fields since several images are needed due to the use of masks. This process can generate images larger than 7000x7000 pixels. Therefore, the computational cost of the technique is high.

Nevertheless, Hough transform is simple to implement and can generate results in real-time, which is highly suitable for clinical applications and completing CTC isolation and counting.

Your comments have been added to the Discussion section.
